# Risk Factors for Failure of Direct Oral Feeding Following a Totally Minimally Invasive Esophagectomy

**DOI:** 10.3390/nu13103616

**Published:** 2021-10-15

**Authors:** Henricus J. B. Janssen, Amaia Gantxegi, Laura F. C. Fransen, Grard A. P. Nieuwenhuijzen, Misha D. P. Luyer

**Affiliations:** 1Department of Surgery, Catharina Hospital, P.O. Box 1350, 5602ZA Eindhoven, The Netherlands; Thijs.janssen@catharinaziekenhuis.nl (H.J.B.J.); laura.fransen@catharinaziekenhuis.nl (L.F.C.F.); grard.nieuwenhuijzen@catharinaziekenhuis.nl (G.A.P.N.); 2Department of Surgery, Vall d’Hebron University Hospital, Universitat Autònoma de Barcelona, 08035 Barcelona, Spain; aganchegui@vhebron.net

**Keywords:** minimally invasive esophagectomy, esophageal cancer, nutrition, enhanced recovery after surgery, jejunostomy

## Abstract

Recently, it has been shown that directly starting oral feeding (DOF) from postoperative day one (POD1) after a totally minimally invasive Ivor-Lewis esophagectomy (MIE-IL) can further improve postoperative outcomes. However, in some patients, tube feeding by a preemptively placed jejunostomy is necessary. This single-center cohort study investigated risk factors associated with failure of DOF in patients that underwent a MIE-IL between October 2015 and April 2021. A total of 165 patients underwent a MIE-IL, in which DOF was implemented in the enhanced recovery after surgery program. Of these, 70.3% (*n* = 116) successfully followed the nutritional protocol. In patients in which tube feeding was needed (29.7%; *n* = 49), female sex (compared to male) (OR 3.5 (95% CI 1.5–8.1)) and higher ASA scores (III + IV versus II) (OR 2.2 (95% CI 1.0–4.8)) were independently associated with failure of DOF for any cause. In case of failure, this was either due to a postoperative complication (*n* = 31, 18.8%) or insufficient caloric intake on POD5 (*n* = 18, 10.9%). In the subgroup of patients with complications, higher ASA scores (OR 2.8 (95% CI 1.2–6.8)) and histological subtypes (squamous-cell carcinoma versus adenocarcinoma and undifferentiated) (OR 5.2 (95% CI 1.8–15.1)) were identified as independent risk factors. In the subgroup of patients with insufficient caloric intake, female sex was identified as a risk factor (OR 5.8 (95% CI 2.0–16.8)). Jejunostomy-related complications occurred in 17 patients (10.3%). In patients with preoperative risk factors, preemptively placing a jejunostomy may be considered to ensure that nutritional goals are met.

## 1. Introduction

Esophagectomy, with or without neoadjuvant treatment, is the cornerstone of curative treatment for esophageal cancer [1,2,3]. Despite the introduction of minimally invasive techniques and enhanced recovery after surgery programs (ERAS), an esophagectomy is still associated with considerable morbidity [4,5,6,7]. Directly starting oral feeding (DOF) is an important element of ERAS and has been shown to be safe and may further improve postoperative outcomes following a (totally) minimally invasive Ivor-Lewis esophagectomy (MIE-IL) [8,9,10].

Insufficient postoperative intake is a major concern in esophageal cancer patients, as dysphagia, weight loss and malnutrition are commonly reported [11,12]. It has been shown that DOF achieves sufficient caloric intake in most patients [9,10] and it does not lead to greater weight loss as compared to tube feeding [9,13]. The main determinants for patients to adhere to the nutritional protocol are complications (e.g., anastomotic leakage) and insufficient intake during the peri-/postoperative phase. Some patients may therefore deviate from this nutritional protocol and still need an artificial nutritional route [9,10].

Two common ways to ensure adequate enteral intake are a nasojejunal tube or a jejunostomy placed during surgery. A jejunostomy is more comfortable for the patient and dislodgment is substantially less when compared with a nasojejunal tube [12,14,15]. However, a jejunostomy may also lead to complications such as catheter occlusion, dislodgement or even bowel obstruction and is reported to be the leading cause of emergency department visits after an esophagectomy [14,16,17]. Ideally, preemptive placement of a jejunostomy should therefore only be performed in patients at risk of inadequate oral intake. Hence, the aim of the study is to identify preoperative risk factors associated with the need for tube feeding after a MIE-IL.

## 2. Materials and Methods

### 2.1. Study Design

This single-center cohort study was conducted at the Catharina Hospital in Eindhoven, the Netherlands, which is a referral hospital for esophageal surgery. All patients aged 18 years or above that underwent an elective MIE-IL, between October 2015 and April 2021, were eligible for inclusion. Only patients with direct start of oral feeding (DOF) from postoperative day one (POD1) as part of the ERAS program were eligible for inclusion, since these patients do not require any tube feeding if sufficient caloric intake is achieved [9,10]. This exact feeding regimen was investigated in the multicenter NUTRIENT II trial in which it was shown that DOF following an MIE-IL is safe and feasible and thereafter became the standard of care at our institution. Patients with the conventional nutritional protocol (i.e., nil-by-mouth and tube feeding for five days postoperatively which was standard of care prior to and during the NUTRIENT II trial) were excluded. There were no additional exclusion criteria.

### 2.2. Perioperative Procedure

A dedicated team of experienced surgeons, anesthesiologists, scrub nurses, intensivists, paramedics, nurse specialists and nurses on the ward were involved in the entire clinical care pathway. The team carries out more than 45 minimally invasive esophagectomies with intrathoracic anastomosis annually [9,10]. Intraoperatively, a low-vacuum Jackson Pratt^®^ drain was placed on the dorsal side of the anastomosis and bilateral chest drains were placed at the end of the procedure. The nasogastric tube was removed at the end of the surgery in all patients. A jejunostomy catheter (9.6Fr) was routinely placed during surgery, but was only used when enteral feeding was required due to a complication, or when caloric intake was insufficient on postoperative day five (POD5).

Patients were extubated directly postoperatively on the OR and were admitted to the ICU overnight. The next day patients were transferred to a specialized surgical unit. Chest X-rays and blood tests were performed postoperatively according to a fixed protocol. Chest drains were removed on day 1. All patients received postoperative care according to a standardized ERAS program [9,10]. Specifically, patients were actively mobilized by the nursing staff to sit in a chair the evening after surgery; on postoperative day 1 (POD1), they started walking and this was gradually expanded to climbing stairs on POD4 and walking 1000 m on POD5. On POD1, all patients started with a liquid oral diet (e.g., soup, yoghurt, porridge, pureed solid foods), supported by oral nutritional supplements (Nutridrink^®^ Compact Protein, Nutricia Advanced Medical Nutrition). This liquid oral diet was gradually expanded until POD15, when a solid diet was initiated. The identical nutritional protocol has previously been shown to be safe and feasible [8,9,10]. A dietician determined daily energy needs for each patient using the Harris–Benedict formula, with a surplus of 30% for energy expenditure in the postoperative phase. On POD5, caloric intake and protein intake were measured and calculated. In patients whose intake did not reach >50% of the calculated energy expenditure on POD5, tube feeding was started using the jejunostomy catheter.

### 2.3. Definitions

Protocol deviation or failure of DOF was defined as any cause that required tube feeding via the jejunostomy. Jejunostomy-related complications were defined as: dislodgement, occlusion, entry site leakage, entry site infection/abscess, intra-abdominal leakage and/or small bowel perforation. Additional postoperative complications were classified according to the Clavien–Dindo classification of surgical complications [18]. Pulmonary complications included: (aspiration)pneumonia, pleural effusion/empyema, pneumothorax and atelectasis requiring intervention, acute respiratory distress syndrome (ARDS) and respiratory insufficiency requiring prolonged treatment or reintubation. Pneumonia was scored using the Uniform Pneumonia Score (UPS) [19]. Anastomotic leakage was scored using the Esophagectomy Complications Consensus Group (ECCG) definition [20].

### 2.4. Statistical Analyses

Analyses were performed with the Statistical Software for the Social Sciences (SPSS) version 25 (IBM Software Group). Categorical data were assessed using the Chi-squared test or the Fisher’s exact test and were presented as an absolute number with the corresponding percentage. After assessment of data normality (assumed normal if both skewness and kurtosis ranges were between −1 to 1), numerical data were analyzed using the independent samples *t*-test or Mann–Whitney U test accordingly. Numerical data are presented as mean with standard deviation (SD) or medians with an interquartile range (IQR). A two-tailed *p*-value of <0.05 was considered statistically significant. To calculate the adjusted estimations of the odds ratio (OR) with the corresponding 95% confidence intervals (CI), statistically significantly and other (clinically) relevant variables (with value *p* < 0.250) were entered into a logistic regression model. In the case of small samples within categorical variables, dichotomous dummy variables were created and then entered in the model. Multicollinearity between categorical variables was assessed using a Chi-squared test for independence and Phi-coefficient of correlation (in case of two categories) or Cramér’s V (for two or more categories).

## 3. Results

In the period between October 2015 and April 2021, 264 patients underwent an elective totally MIE-IL, at the Catharina Hospital Eindhoven. In 99 patients, the feeding regimen—as previously described in the methods—was nil-by-mouth and tube feeding for five days postoperatively and these patients were therefore excluded. In 165 patients, the feeding regimen was DOF on POD1 and all 165 eligible patients were included in the analyses. Of note, 85 of these patients participated in the NUTRIENT II trial and a prospective cohort study conducted at our institution [9,10].

### 3.1. Baseline Characteristics

Mean age was 65 years (SD ± 9) and patients were predominantly male (79.5%). Median body mass index (BMI) was 25.9 kg/m^2^ (IQR 23.5–29.2). Comorbidities were present in 106 patients (64.2%). Most patients (69.7%) were American Society of Anesthesiologists (ASA) classification II. Adenocarcinoma (AC) was the predominant histological subtype (86.7%). A total of 155 patients (93.9%) underwent neoadjuvant treatment (routinely chemoradiotherapy). Forty-one patients (24.8%) required preoperative tube feeding due to insufficient oral intake or inability of oral intake (e.g., dysphagia). Preoperative baseline characteristics are further subdivided by adherence to the nutritional protocol and are shown in Table 1.

### 3.2. Postoperative Outcomes

The thirty-day overall postoperative complication rate was 53.3% (*n* = 88). Cardiopulmonary complication rate was 43.0%. More specifically, a cardiac complication occurred in 33 patients (20.0%). Atrial fibrillation was the most frequently diagnosed cardiac complication (*n* = 29). A pulmonary complication occurred in 54 patients (32.7%) and the pneumonia rate (UPS) was 21.8% (*n* = 36). Anastomotic leakage (AL) was diagnosed in 15 patients (9.1%). Of these, seven were minor leaks treated conservatively with antibiotics, a nasogastric tube with suction and dietary modification (ECCG Type I). Six required endoscopic stenting (ECCG Type II) and two required surgical (re)intervention (ECCG Type III). Chyle leakage occurred in 5 patients (3.0%). Gastric conduit distention was diagnosed in eight (4.8%) patients. Jejunostomy-related complications occurred in 10.3% of patients (*n* = 17). In ten cases the jejunostomy catheter dislodged; in three cases an occlusion of the catheter; and in four entry site infections occurred. Eight patients required an extra visit to the outpatient clinic or emergency department for removal or (radiological) replacement. One of these patients was readmitted. In the patients that successfully followed the intended nutritional protocol (*n* = 116), nine (7.8%) had a jejunostomy-related complication, whilst the jejunostomy was not used.

Median length of hospital stay was 7 days (IQR 6–10) and hospital readmission rate was 10.3% (*n* = 17). Of the patients that were readmitted, 11 patients (6.7%) had a late surgical complication, i.e., anastomotic leakage (*n* = 3), pulmonary complication (*n* = 3), pain complaints (*n* = 3), dislodged jejunostomy catheter (*n* = 1) or occult bacteremia requiring antibiotics (*n* = 1). Four patients were shortly readmitted to initiate tube feeding due to insufficient caloric intake as a result of functional feeding-related symptoms (*n* = 3) or a pyloric spasm that required endoscopic dilatation (*n* = 1). Three patients were readmitted for other, non-surgical reasons.

### 3.3. Nutritional Protocol Deviation

A total of 49 patients (29.7%) deviated from the nutritional protocol and required tube feeding postoperatively. In these patients, pulmonary complication rate was significantly higher compared to patients that did not deviate from the protocol (respectively, 59.2% versus 21.6%, *p* < 0.001). More specifically, pneumonia rate (UPS) was 40.8% in patients that deviated from the protocol versus 13.8% in patients without a protocol deviation (*p* < 0.001). Evidently, patients with AL and patients with gastric conduit distension deferred from oral intake and required tube feeding, thereby deviating from the nutritional protocol. An univariable regression analysis was performed (Table 2) and showed that sex (female) and histological subtype (SCC) were negatively associated with nutritional protocol deviation for any cause (OR 3.1 (95% CI 1.4–6.8) and OR 2.7 (95% CI 1.1–7.0), respectively). On the other hand, a significant protective association between higher (clinical) tumor stage (cTNM III + IV versus cTNM 0-II) was observed (OR 0.5 (95% CI 0.2–0.9)), whereas this association was not identified for pathological tumor stage ((y)pTNM III + IV versus (y)pTNM 0-II); OR 0.977 (95% CI 0.5–1.9). Considering the number of events per variable, the multivariable analysis was performed with sex, ASA score and histological subtype, as these were deemed most (clinically) relevant. Subsequently, the multivariable analysis retained sex (female) as an independent factor associated with nutritional protocol deviation for any cause (OR 3.5 (95% CI 1.5–8.1)), whereas the association of histological subtype (SCC) was not statistically significant (*p* = 0.091). Moreover, higher ASA scores (III + IV) were associated with nutritional protocol deviation for any cause (OR 2.2 (95% CI 1.0–4.8)). A multicollinearity analysis was performed using the chi-square test, which showed a significant association between sex and ASA score (*p* = 0.004) and histology (*p* = 0.022), but not between ASA score and histology (*p* = 0.370). Phi-coefficient was calculated for these dichotomous (categorical) variables and showed a weak correlation between sex and ASA score (φ −0.222), and sex and histological subtype (φ 0.178).

In addition, a subgroup analysis was performed to discriminate between patients that deviated from the protocol due to a complication and patients without complications directly leading to protocol deviation. In the 31 patients that required tube feeding, the cause was a complication prohibiting or hindering oral intake (Appendix A). AL was the primary reason in 15 patients, whereas in nine patients this was due to respiratory insufficiency. In four patients, gastric conduit distension was the primary reason for protocol deviation. The remaining three patients deviated from the protocol due to chyle leakage (*n* = 1), pyloric spasm (*n* = 1) and an intraoperative complication (*n* = 1). In this subgroup, higher ASA scores (OR 2.8 (95% CI 1.2–6.8)) and SCC (OR 5.2 (95% CI 1.8–15.1)) were found to be independent risk factors for complications leading to protocol deviation on multivariable analysis (Table 3).

On the contrary, 18 of 49 patients required tube feeding due to insufficient oral intake (Appendix A). In this subgroup, 13 patients did not reach sufficient caloric intake on POD5—as previously defined—and five patients had passage/swallowing problems postoperatively in the absence of abnormalities on endoscopy. In this group, univariable analysis showed that female sex was a risk factor for protocol deviation due to insufficient oral intake (OR 5.8 (95% CI 2.0–16.8)).

## 4. Discussion

This study identified several risk factors associated with deviation from the nutritional protocol (directly starting oral intake) for any cause after a MIE-IL. Previously, it was shown that the main postoperative determinants for patients to adhere to this nutritional protocol postoperatively are complications (e.g., AL and pneumonia) or insufficient caloric intake [8,9,10,21]. To our knowledge, this is the first study that aimed to investigate preoperative risk factors in this patient group (patients directly starting with oral intake after a MIE-IL). Several preoperative patient and histopathologic factors (i.e., female sex, SCC and higher ASA scores) were independently associated with failure of DOF.

In patients with complications hindering or prohibiting oral intake, AL and respiratory insufficiency were the most frequently observed complications that led to failure of DOF [8,9,10,21]. Several risk factors (e.g., older age, higher ASA scores, obesity, diabetes and smoking) that predispose for AL and respiratory complications in esophagectomy patients have been identified in previous studies [22,23,24,25,26]. Due to the limited amount of patients in this cohort, preoperative risk factors for these complications were not analyzed individually, but as one subgroup. In this subgroup, higher ASA scores and SCC were retained as independent risk factors for any postoperative complication leading to failure of DOF on multivariable analysis. This has not been described before in patients following a MIE-IL. Considering that higher ASA scores are also a risk factor for postoperative morbidity, especially AL and pneumonia (also shown in Appendix A), we hypothesize that these factors are correlated and are therefore also associated with the failure of DOF [22,25].

For squamous cell carcinoma, this may be explained by its underlying pathogenesis, wherein major etiological factors are tobacco smoking and alcohol consumption—which also act synergistically—and poor diet [27]. It has previously been reported that these patients have a higher preoperative risk profile, e.g., due to a poorer nutritional status and poorer pulmonary function, thereby increasing the risk of postoperative morbidity in these patients [27,28,29]. Although data on the extent of smoking (pack years) and the exact amount of alcohol consumption in this current cohort was limited, there was no significant association between SCC and a higher preoperative risk profile (i.e., smoking, alcohol consumption, obesity, comorbidity), as shown in Appendix A. On the other hand, patients with SCC at our institution routinely undergo an extended two-field lymph node dissection (2FLND) as compared to a standard 2FLND in patients with AC. In these patients, lymph nodes along the aortopulmonary window and paratracheally are carefully harvested, while preserving the vagus and recurrent nerves. Earlier studies have shown that the extent of lymph node dissection is associated with higher AL rates, which may explain the higher incidence of AL in patients with SCC that underwent an extended 2FLND [30,31,32]. However, these studies compared 2FLND with three-field lymph node dissections; thus, other surgical factors (e.g., anastomotic technique and location of anastomosis) were potential confounders. Moreover, lymph node yield did not significantly differ between patients squamous cell carcinoma (SCC) and adenocarcinoma (AC) (median 29 (IQR 21–33) vs. median 27 (IQR 21–32), *p* = 0.740) in the current cohort. The association between histological subtype and occurrence of AL after an esophagectomy has not been extensively researched [33,34]. One retrospective study by Herzberg et al. also identified SCC as a risk factor for AL; however, there may have been potential confounders in their study, such as different types of surgery [33]. On the contrary, in a population-based study by Rutegård et al., no significant difference in AL rate was observed between SCC and AC [34]. More data are needed to further substantiate this finding.

Next, patients may also deviate from the intended nutritional protocol due to insufficient caloric intake. This finding was in line with previous studies on DOF, following a MIE-IL and may be explained by (functional) feeding related symptoms that occur shortly after surgery (e.g., nausea, vomiting, early satiety and dysphagia) [8,9,10]. Interestingly, in this subgroup, only female sex was identified as a risk factor for insufficient oral intake. This is a new finding and has not been described before in patients undergoing an MIE-IL. However, previous studies in other upper gastrointestinal surgery (e.g., pancreaticoduodenectomy, gastrectomy and bariatric surgery) [35,36,37], as well as in colorectal surgery [38], reported that females tend to have lower total dietary intake following surgery. It was suggested that this may be due to a slower recovery in food tolerance after reintroducing oral diet and because female patients tend to more gradually and carefully increase the food amount postoperatively. In patients undergoing an esophagectomy as well as a gastrectomy for cancer, it has been reported that female sex is associated with an increased risk of postoperative weight, but data are inconclusive [39,40].

Most patients (70%) in the current study did not deviate from the intended nutritional protocol and therefore did not need the jejunostomy. Preemptive placement of a jejunostomy may confer an unnecessary risk in the postoperative period in these patients, since tube-related complications (e.g., dislocation, occlusion, leakage, entry site infections, and GI complaints) are well-known and can impact patients’ recovery and quality of life [12,41,42,43]. In the current study, jejunostomy-related complications occurred in 10% of patients, which is less frequent than what is reported in the majority of previous studies [12,14,15]. These were mostly minor complications, e.g., dislodgement, dysfunction/occlusion and local entry site inflammation/infection. On the other hand, 30% of patients (*n* = 49) required tube feeding. Hence—in case no jejunostomy was placed during the initial surgery—secondary placement of a jejunostomy (or nasojejunal tube) would have been necessary in these patients, which also confers a risk. Although these data could aid surgeons in making more appropriate treatment decisions, the relatively small sample size of the current cohort and different underlying causes (complications versus no complications leading to failure of DOF), limit the decision-making to omit preemptive placement of a jejunostomy. Moreover, due to the specificity of this cohort (patients with DOF following a MIE-IL), these data may not be applicable to institutions with different feeding regimens and/or techniques and should therefore be substantiated in future studies.

Furthermore, this study was limited due to the retrospective design and by the fact that only a single center was included. Nevertheless, all eligible patients were included, thereby limiting potential selection bias. Nevertheless, it might aid surgeons in selecting better and more appropriate treatment decisions. The strength of this study is that patients were included from a high-volume center, with a relatively low, stable complication rate and perioperative conditions; e.g., MIE with minimal learning associated morbidity [44,45] and a well-established ERAS program [9,10] were standardized in all patients. This is essential since the risk for failure of DOF is higher in a setting of a high complication rate and a non-standardized postoperative protocol (e.g., AL and respiratory complications). Consequently, length of hospital stay was also shorter than previous reports [46,47,48,49]. This was accompanied by a total readmission rate of 10%, which is in line with previous nationwide studies [46,47,48], which reported a readmission rate ranging from 7.0–15% and below the 15% reported in a Dutch national cohort study [49].

## 5. Conclusions

In conclusion, this study identified preoperative risk factors associated with the failure of oral feeding directly postoperatively in patients after an MIE-IL. These data may aid surgeons in their decision-making for preemptive placement of a jejunostomy or nasojejunal tube in patients at risk, to ensure adequate caloric intake postoperatively.

## Figures and Tables

**Table 1 nutrients-13-03616-t001:** Preoperative patient baseline characteristics by adherence to feeding regimen.

	No Deviation		Deviation from Protocol		
	(*n* = 116)		(*n* = 49)		*p*-Value
**Age, years**	65	(9)	64	(8)	0.600
**Sex, male**	99	(85.3)	32	(65.3)	0.004
**BMI, kg/m^2^**	26.1	(23.4–29.3)	25.0	(23.5–29.2)	0.651
**Weight loss, kg**	0	(0–5)	3	(0–7)	0.402
**Preoperative tube feeding**	25	(21.6)	16	(32.7)	0.132
**Smoking history**	92	(79.3)	40	(81.6)	0.733
Active smoker (or quit <1 year)	33	(28.7)	17	(34.7)	0.445
**Alcohol consumption**					0.552
Daily	38	(33.0)	15	(30.6)	
Weekly	18	(15.7)	5	(10.2)	
**Comorbidity**	74	(63.8)	32	(65.3)	0.853
Cardiac	17	(14.7)	8	(16.3)	0.784
Pulmonary	23	(19.8)	11	(22.4)	0.704
Vascular	40	(34.5)	17	(34.7)	0.979
Diabetes	15	(12.9)	4	(8.2)	0.381
Obesity	25	(21.6)	10	(20.4)	0.870
**ASA Class**					0.189
II	86	(74.1)	32	(65.3)	
III	27	(23.3)	17	(34.7)	
IV	3	(2.6)	0		
**Histology**					0.082
Adenocarcinoma	105	(90.5)	38	(77.6)	
Squamous cell carcinoma	10	(8.6)	10	(20.4)	
Undifferentiated	1	(0.9)	1	(2.0)	
**(c)TNM stage**					0.018
0	1	(0.9)	1	(2.0)	
I	4	(3.4)	9	(18.4)	
II	22	(19.0)	9	(18.4)	
III	61	(52.6)	23	(46.9)	
IV	28	(24.1)	7	(14.3)	

**Legend:** values are absolute numbers (percentage) or medians (lower quartile—upper quartile) or means (standard deviation). BMI, body mass index; ASA, American Society of Anesthesiologists; (c) TNM, clinical tumor stage.

**Table 2 nutrients-13-03616-t002:** Univariable and multivariable regression for nutritional protocol deviation for any cause.

	OR	95% CI	*p*-Value
** Univariable **			
Sex (female vs. male)	3.1	(1.4–6.8)	0.005
ASA Score (III + IV vs. II)	1.5	(0.7–3.1)	0.252
Histology (SCC vs. AC + undifferentiated)	2.7	(1.1–7.0)	0.039
Tumor stage (cTNM III + IV vs. 0-II)	0.5	(0.2–0.9)	0.033
Preoperative tube feeding (yes vs. no)	1.8	(0.8–3.7)	0.134
** Multivariable **			
Sex (female vs. male)	3.5	(1.5–8.1)	0.004
ASA Score (III + IV vs. II)	2.2	(1.0–4.8)	0.048
Histology (SCC vs. AC + undifferentiated)	2.4	(0.9–6.4)	0.091

**Legend:** OR, odds ratio; CI, confidence interval; ASA, American Society of Anesthesiologists; SCC, squamous cell carcinoma; AC, adenocarcinoma; (c) TNM, clinical tumor stage.

**Table 3 nutrients-13-03616-t003:** Univariable and multivariable regression for nutritional protocol deviation (complication subgroup).

	OR	95% CI	*p*-Value
** Univariable **			
Sex (female vs. male)	2.0	(0.8–5.3)	0.147
ASA Score (III + IV vs. II)	2.4	(1.0–5.4)	0.040
Histology (SCC vs. AC + undifferentiated)	4.3	(1.6–12.0)	0.004
Tumor stage (cTNM III + IV vs. 0-II)	0.4	(0.2–0.9)	0.032
Preoperative tube feeding (yes vs. no)	1.7	(0.7–4.2)	0.217
** Multivariable **			
ASA Score (III + IV vs. II)	2.8	(1.2–6.8)	0.019
Histology (SCC vs. AC + undifferentiated)	5.2	(1.8–15.1)	0.002

**Legend:** OR, odds ratio; CI, confidence interval; ASA, American Society of Anesthesiologists; SCC, squamous cell carcinoma; AC, adenocarcinoma; (c) TNM, clinical tumor stage.

## Data Availability

Data (anonymized) will be made available on request, such as data on patient characteristics and postoperative outcomes to reproduce the analyses or to gain an insight into how definitions were scored. A request can be send to the corresponding author from the publication date to one year after the publication date (since data are only stored for this period, as approved by the review board).

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
