# Peer review of "Risk Factors for Failure of Direct Oral Feeding Following a Totally Minimally Invasive Esophagectomy"

_nutrients, 2021, doi:10.3390/nu13103616_

Round 1

Reviewer 1 Report

The authors did a very important study looking at risk factors associated with the need for tube feeding after esophagectomy. This is a very nice and important study, well written and provides lots of information.

From my point of view the main question could be answered in more detail. The question regarding risk factors and odds ratios for these factors could be shown more clearly, maybe in a graph. Also in the discussion the risk factors should be discussed in more detail and more precise.

Another interesting point is why in your study type of smoking and alcohol consumption was no risk factor. This could be discussed in more detail.

Also please explain why in your study data regarding smoking and alcohol consumption was limited.

The argumentation why female patients tend to have a lower total dietary intake following surgery is rather weak and should be rewritten. What were the reasons provided in other studies? As nausea is more common in female patients after surgery, was there a positive association?

Author Response

We thank the reviewer for the interest in and critical appraisal of our study. The reviewer suggested to discuss the main question regarding the risk factors in more detail and provide a graphical overview of the odds ratios. The discussion has been revised and the risk factors are described in more detail, also including the additional analyses as suggested by another reviewer. Odds ratios from the univariable analyses are presented in Table 2, but due to the correlation of these risk factors (supplementary table 1), a multivariable analysis was not feasible and therefore a graphical overview of the odds ratios was not practicable.

Furthermore, the reviewer rightly mentions that it is unexpected that smoking and alcohol consumption were no risk factors and also requested clarification on why data regarding smoking and alcohol consumption were limited. However, data on smoking history and alcohol consumption were essentially available in all patients as this is routinely asked at the patient intake, but this is frequently limited to a yes/no question instead of the amount as estimated by the patient. In our experience, the attending physicians are more interested in whether patients are actively smoking and/or in (daily) alcohol consumption. In such cases patients are encouraged and assisted to quit and, if not achievable, substantially lessening tobacco and/or alcohol abuse during the preoperative phase. Nonetheless, we agree with the reviewer that this limited the ability to detect the association between heavy smoking and/or drinking with nutritional protocol deviation and postoperative morbidity.

In the current study active smokers were both patients that were smoking at the time of diagnosis/patient intake and patients that quit less than 1 year prior to patient intake, since this has been shown to still increase the risk of postoperative morbidity (Takeuchi H et al, Ann Surg. 2014; Yoshida et al. World J Surg, 2016). The reason that smoking and alcohol consumption were not identified as risk factors for nutritional protocol deviation, may be due to the post-cessation stimulating effect on appetite and food intake that can last even months after smoking cessation (Bush T et al. Obesity, 2016). Contrarily, alcohol consumption has been suggested to stimulate and potentially increase food intake, but these effects are mostly short-term when alcohol is consumed (Kwok et al. Br J Nutr, 2019) and may therefore not affect postoperative appetite and food intake.

On the other hand, in the sub analysis (Supplementary File I, tables V and VI) it was shown that (active) smoking and (daily) alcohol consumption were associated with the occurrence of pulmonary complications, especially pneumonia. Furthermore, smoking history was associated with the occurrence of anastomotic leakage (AL) and there was a trend towards active smoking and the occurrence of AL, but not (daily) alcohol consumption. These findings were in line with previous reports from several meta-analyses (Shabanzadeh et al. J Gastrointest Surg 2014; Van Daele. Interact Cardiovasc Thorac Surg, 2016; Eliasen et al. Ann Surg, 2013).

Lastly, the reviewer debated that the argumentation for female gender as a risk factor for insufficient postoperative intake was weak and requested more discussion on this. The reviewer brings up a good point, however, the underlying mechanism is not well studied and we compared our data with other studies in upper gastrointestinal surgery (e.g. pancreaticoduodenectomy, gastrectomy and bariatric surgery) as well as colorectal surgery. In these studies it was reported that females tend to have lower total dietary intake following surgery (e.g. due to a slower recovery in food tolerance after reintroducing oral diet and because female patients tend to more gradually and carefully increase the food amount postoperatively). As previously described in the manuscript, in the current study 13 of 18 patients did not reach sufficient caloric intake (at least 50% of calculated energy expenditure) on postoperative day 5 due to a lack of appetite, early satiety and/or nausea, whereas 5 patients had passage/swallowing problems in absence of endoscopic findings. These findings were also observed more frequently in female patients.

Reviewer 2 Report

The authors reported their retrospective cohort study to identify the preoperative risk factors for failure of direct oral feeding protocol following MIE.

Though the intention of the analysis might be original and valuable, the methodology was not good.

Because it is impossible to predict the occurrence of either postoperative complications or insufficient intake, the main outcomes should be analyzed for the deviation for any reason. Analyses for complications and insufficient intake should be done as sub-analyses. Rather than analyzing for what they mentioned as the primary aim of the study to identifying preoperative risk factors, they ended with analyzing the cause of deviation by dividing them into postoperative complications and insufficient intake.

Tables were unintelligible without proper headings and alignments.

Surgical quality, especially the intensity of the “extended 2-filed” was not evident in the manuscript. Any difference in lymph node yields? How was the incidence of recurrent laryngeal nerve palsy, which should have an impact on oral intake? What was the definition of MIE-IL? Were all of them operated by totally minimally invasive, or hybrid?

Author Response

We thank the reviewer for the interest and critical appraisal of our manuscript. The reviewer rightly debated that the analysis of the primary aim was missing and should include deviation for any cause and that analyses for complications and insufficient intake should be done as sub-analyses. These analyses have been added to the manuscript. Furthermore, the reviewer pointed out that tables were unintelligible and we agree. To include the additional analyses and present these data more clearly, all tables have been altered. The previous table 1 showed baseline patient characteristics for the entire cohort and this has been removed, whereas relevant data from the entire cohort are now presented only in the text. In the revised Table 1, baseline patient characteristics are now subdivided in patient without nutritional protocol deviation and patients with protocol deviation. Risk factors and odds ratios for this group are presented in Table 2. Due to the correlation of the risk factors identified on univariable analysis (as shown in Supplementary Table 1), a multivariable analysis was not feasible for the primary outcome (protocol deviation for any cause).

Furthermore, the reviewer raised a point that surgical quality (i.e. intensity of the “extended 2-field” and lymph node yield, laryngeal nerve palsy and type of minimally invasive surgery) was not evident in the manuscript. This has been described in more detail and we thank the reviewer for pointing this out. Firstly, all patients underwent a totally minimally invasive Ivor-Lewis esophagectomy, although in one patient conversion to laparotomy was necessary due to an intraoperative complication. Secondly, in patients with squamous cell carcinoma or preoperatively diagnosed pathological lymph nodes at this site, lymph nodes along the aortopulmonary window and paratracheally are routinely included with the 2-field lymphadectomy, while carefully preserving the vagus and recurrent nerves. However, lymph node yield did not significantly differ between patients squamous cell carcinoma (SCC) and adenocarcinoma (AC) (median 29 [IQR 21-33] vs. median 27 [IQR 21-32], p=0.740). Similarly, lymph node yield did not significantly differ between patients with anastomotic leakage (AL) and patients without anastomotic leakage (median 29 [IQR 20-32] vs. median 27 [IQR 22-33], p=0.986). The association between histological subtype and occurrence of anastomotic leakage after an esophagectomy has not been extensively researched. One other retrospective study identified squamous cell carcinoma as a risk factor for AL (Herzberg et al. Langenbecks Arch Surg, 2021). However, there may have been potential confounders such as type of surgery. On the contrary, in a population-based study by Rutegård et al (Ann Surg Oncol, 2012) no significant difference in AL rate was observed between SCC and AC. More data are still needed to further substantiate this finding.

The reviewer also enquires about the incidence of laryngeal nerve palsy, however, this was not diagnosed in any patient. 

Reviewer 3 Report

As in many Asian centers complete esophagectomy with a cervical anastomosis are performed, it would be interesting to know the experiences of the authors with this particular patient group.

Basically, most findings are already known, and it remains unclear whether female patients need another treatment approach.

Unfortunately, almost have of the patients have already been included and published. 

15 days until a normal diet was introduced seems to be very long. What are the considerations for such a long dietary restriction? 

Author Response

We thank the reviewer for the interest in and critical appraisal of our study. The reviewer asks for our experience in patients after an esophagectomy with a cervical anastomosis (McKeown). Unfortunately we cannot comment on this since we focused on minimally invasive Ivor-Lewis esophagectomy (MIE-IL) with linear stapled side-to-side anastomosis that has been the most performed procedure at our institution since 2016. In the investigated time period, only four patients underwent a minimally invasive McKeown esophagectomy and received direct oral feeding (DOF) postoperatively, but these patients were not included in this study.

The reviewer mentions that most findings are already known and it remains unclear whether female patients need another treatment approach. However, we do not agree with this since the aim of this study, as previously described, was to identify preoperative risk factors associated with failure of this nutritional protocol (DOF) in this specific patient population (MIE-IL directly starting oral intake postoperatively). Hence, all limited available data (including data from previously included patients) were needed.

Lastly, the reviewer enquires about considerations for the dietary restriction of 14 days before a normal diet with solid foods is initiated. We thank the reviewer for raising this interesting point. This semi-liquid feeding regimen and enhanced recovery after surgery (ERAS) program was investigated in the NUTRIENT II trial and prospective cohort study and thereafter became the standard of care without alterations. The 14-day period for solid foods was chosen at the time, taking into account the healing process of the esophageal anastomosis. This process is relatively similar to other tissues (within the 2nd to the 14th postoperative day definite wound closure begins) and it has been reported in research studies that the maximal strength is approached at approximately 14 days (Rijcken et al. J Surg Res, 2014; Dominguez et al. J Surg, 1996). We hypothesized that with solid foods (e.g. bread/dough balls/meat) tension on the anastomosis may occur and/or dislocate sutures that could hinder this healing process. Liquid or pureed foods are generally also preferred since gastric emptying is often preserved for softer consistencies which decreases the likelihood for retention of food in the gastric conduit (Lee et al. Ann Thorac Surg, 2005; Ukegjini et al. Langenbecks Arch Surg, 2021). In addition, solid foods are reported to have a greater satiating effect (Tieken et al. Horm Metab Res, 2007  StribiÅ£caia et al. Sci Rep, 2020). Hence, we believe (semi)-liquid or pureed foods make it easier for patients’ to adhere to our nutritional protocol and reach nutritional goals while they are recovering from surgery.

Round 2

Reviewer 2 Report

I disagree with the statistical analyses, especially regarding the multivariate analysis having not been conducted regarding table2. Supplemental table I means nothing.

Unless there is the proper support of statisticians, I cannot accept their analyses.

“grade” is wrongly used and confusing. It should be stage instead.

Author Response

We are grateful to the reviewer for the interest in our study and appreciate the time and effort dedicated to providing us with insightful feedback on the manuscript. The reviewer did not agree with the statistical analyses, especially regarding the multivariate analysis that was not conducted regarding Table 2. The reviewer was correct that there was a flaw in the analyses and we have consulted the senior epidemiologist at our institution (Catharina Hospital) to verify the statistical analyses.

The previous Supplementary Table I was incorrectly interpreted as part of the multicollinearity analysis and consequently the analyses were incomplete. Although the chi-squared test according to this contingency table was correct, it is only a test of independency between variables and not of correlation. On the basis of a significant association between variables, it was therefore incorrectly assumed that categorical variables were also correlated. The analyses have now been revised and the correlation as calculated by the Phi-coefficient (as variables were dichotomous) has now been presented. In addition, the multivariable analysis has been performed since the correlation between variables was only weak. Supplementary Table I has been deleted as suggested by the senior epidemiologist, as this was redundant and should be described in the methods.

This manuscript is a resubmission of an earlier submission. The following is a list of the peer review reports and author responses from that submission.

Round 1

Reviewer 1 Report

This article provides the information about the risk factor of deviation from oral intake protocol, which the author’s group has been using. This issue is very interesting, though those fact has been already published by the same author’s group (Ann Surg 2020). Although this article increased the number of patients for evaluation, there was not much new information in this article. To improve this article, the author should precisely assess the matter of insufficient oral intake from their result and evaluate the solution by each.

In detail, it was nothing surprise that the anastomosis leakage and pneumonia are the reasons for hindering DOF. Why the patient with SCC had the risk of insufficient oral intake? The author just mentioned that they had the risk for postoperative complication, but that would not tell the reason for this result. The author must evaluate what kind of complication has affected this result, and that would be the answer for the risk of insufficient oral intake.

Author Response

Reply to comment 1:

Indeed, our research group has previously reported on the postoperative nutritional outcomes after a minimally invasive Ivor-Lewis esophagectomy at our institution. However, after conducting these trials it came to our attention that even though most patients had sufficient caloric intake – and therefore did not need the pre-emptively placed jejunostomy – approximately 30% required tube feeding by the. The aim of the current study therefore was to identify the underlying (risk) factors that could explain this and potentially limit the placement of a jejunostomy by preoperatively selecting patients at risk.

Reply to comment 2:

As discussed in the manuscript, main determinants for patients to adhere to the nutritional protocol during the peri-/postoperative phase are indeed complications (anastomotic leakage, respiratory insufficiency and gastric conduit distension). We also identified a higher ASA grade and squamous cell carcinoma as risk factors for protocol deviation. Since this is the first study to report on these risk factors these data could not be compared to the literature which limits the discussion. However, given the underlying etiology, we hypothesized that risk factors for overall postoperative complications are also associated with nutritional protocol deviation. The increased risk of postoperative complication in higher ASA grades is well-known, however, for squamous cell the underlying etiology is less clear. We thank the reviewer for this comment and have added more discussion on this topic (page 7).

Reviewer 2 Report

Congratulations to a nicely organized manuscript and pioneer work using ERAS pathways in MIS esophageal surgery.

While median LOS of 7 days is wonderful, 10% readmission rate would not be acceptable in some countries. Please add a paragraph to the discussion commenting this finding.

Avoid judgemental comments in the methods (" ..since it was previously shown that this optimized ERAS program improved recovery and outcomes for patients undergoing a MIE-IL") as some of these, even if you have a reference, may be highly debatable and should be part of the discussion.

What is the justification for a jejunostomy if 10% of them have complications? I really doubt this is a helpful add-on to this complex surgery. This problem has been well adressed in the discussion but I wonder if any conclusions were taken by the authors. Is this still standard of care at their institution? Please discussed with more detail- I suggest adding to the conclusion that a preemptive jejunostomy should not be performed in low risk patients.

Author Response

Replies to reviewer 2:

We thank the reviewer for the interest in our study and congratulating us on our work in this field. Moreover, we thank the reviewer for critically appraising our study. Our comments are summarized below:

Reply to comment 1:

This is an interesting point and we agree with the reviewer that the hospital readmission rate needs to be taken into account. This re-admission rate however, is in line with rates reported in the literature from nationwide cohort studies ranging from 7.0%-15% (Yerokun et al, Ann Thorac Surg 2016; Kauppila et al, Ann Surg Oncol 2018; Thirunavukarasu et al, Int J Surg 2016; van der Werf et al, Ann Surg 2020). We have added this to the discussion (page 7). Furthermore, details were lacking and we have added more information on the reasons for readmitting patients to the results section (page 5), as this includes all readmissions regardless of underlying reason and not only (late) surgical complications. In 7.3% of patients (n=11) these were (late) surgical complications i.e. anastomotic leakage (n=3), pulmonary complication (n=3), pain complaints (n=3), dislodged jejunostomy catheter (n=1) or occult bacteremia requiring antibiotics (n=1). Four patients were shortly readmitted to initiate tube feeding due to insufficient caloric intake as a result of functional feeding related symptoms (n=3) and pyloric spasm that required endoscopic dilatation (n=1). Two patients were readmitted for other, non-surgical reasons.

Reply to comment 2:

We thank the reviewer for pointing this out to us. Indeed, the improved postoperative outcomes may not be applicable to all esophagectomy patients and we have corrected this in the manuscript. It should now still clearly indicate why we selected only patients that directly started with oral feeding postoperatively (page 2).

Reply to comment 3:

This is also an interesting remark and aligns with the aim of our study. Indeed, preemptive placement of a jejunostomy was standardly performed at our institution and this study aimed to select patients for jejunostomy placement rather than performing this as standard of care. In our cohort, 30% of patients required tube feeding. In these patients we prefer a jejunostomy over a nasojejunal tube since it is more comfortable for the patient and dislodgement is substantially less when compared with a nasojejunal tube (Berkelmans et al, J Thorac Dis 2017). We believe the data in the current study might aid surgeons in making more appropriate treatment decisions, but due to the relatively small sample size and different underlying causes (complications versus no complications leading to nutritional protocol deviation), this should be assessed further to better select patients at risk. We have also added more discussion on this in the main text (page 7).

Round 2

Reviewer 1 Report

The author mentioned that patient with SCC tend to have the habit of smoking and alcohol which lead to poor diet after surgery. Then, author should seek this result by the habit of smoking, alcohol from their cohort. SCC itself would not cause poor oral intake. So the author should evaluate from their underlying patient's condition, if the author intended to  identify the risk factors of insuffcient oral intake and potentially limit the placement of a jejunostomy by preoperatively selecting patients at risk .

Author Response

We agree with the reviewer that it is not likely that SCC itself would cause insufficient oral intake/protocol deviation. However, what we observed was that more patients with SCC had a complication leading to protocol deviation as compared to adenocarcinoma. This is an important distinction because main determinants for patients to adhere to the nutritional protocol are both insufficient caloric intake due to complications (e.g. anastomotic leakage, respiratory insufficiency) and insufficient caloric intake due to other factors than complications (functional complaints; e.g. dysphagia, lack of appetite, fear of eating, etc.).

As previously mentioned, since this is the first study to report on risk factors for failure of DOF we could not compare our data to the literature. However, it has previously been reported that SCC is a risk factor for postoperative morbidity due to the higher preoperative risk profile (Gockel et al, World J Surg Oncol 2005; Kamarajah et al, Ann Surg Oncol 2021; Yoshida et al, World J Surg 2018). Since complications are main determinants for protocol deviation we hypothesized that SCC (and ASA) are therefore also associated with protocol deviation.

Whether patients with SCC had a higher preoperative risk profile in the current study could not be determined, since the extent of smoking (packyears) was only available in 61 of 120 (50.8%) of patients with a smoking history. Moreover, it was not reported in the patients’ records if and when current smokers quit during the preoperative phase. Similarly, it was only reported if patients used alcohol and how often. Data on the amount of alcohol consumption were unavailable. We have added this limitation to the discussion (page 6) and added alcohol consumption in Table 2. This should be assessed further in new studies as previously mentioned in the manuscript.